# Characteristics of Aerosol Extinction Hygroscopic Growth in the Typical Coastal City of Qingdao, China

Nana Liu [1,2], Shengcheng Cui [1,2,3,*], Tao Luo [1,2,3], Shunping Chen [1,4], Kaixuan Yang [1,4], Xuebin Ma [1,4], Gang Sun [1,2] and Xuebin Li [1,2,3]

1  Key Laboratory of Atmospheric Optics, Anhui Institute of Optics and Fine Mechanics, HFIPS, Chinese Academy of Sciences, Hefei 230031, China
2  Advanced Laser Technology Laboratory of Anhui Province, Hefei 230037, China
3  Nanhu Laser Laboratory, National University of Defense Technology, Changsha 410073, China
4  Science Island Branch of Graduate School, University of Science and Technology of China, Hefei 230026, China
*  Correspondence: csc@aiofm.ac.cn

**Abstract:** The aerosol hygroscopic growth (HG) characteristics in coastal areas are very complex, which is one of the main influences on the simulation accuracy of radiation transfer modeling for coastal environments. Previous studies have shown that aerosol HG characteristics are very different in open oceans and inland regions. However, the aerosol HG features in coastal areas are strongly affected by its type. In this work, an aerosol backward trajectory tracing model was used to classify the local aerosol type. Using long-term field campaign data in Qingdao (25 September 2019 to 25 October 2020), the HG characteristics of different types of aerosols (i.e., land source, sea source, and mixed aerosol) under different seasons and different atmospheric environments (i.e., pollution background and clean background) were studied. Quantitative models of aerosol HG factor were established for aerosols from different sources in different seasons and under different pollution background conditions. The major type of local aerosol is terrestrial aerosol, as the marine source only accounts for 10–20%. Seasonal HG characteristics (deliquescence point, DP) of mixed and land source aerosol vary significantly, from around RH = 60% to RH = 85%, while that of the marine aerosol is rather consistent (RH = 80%). When the atmospheric background is relatively clean, the DPs of aerosols from different sources are almost the same (about RH = 80%), but when the pollution is heavy, the DPs of terrestrial aerosols are almost 20% lower than those of marine sources. These models can be directly used to characterize the hygroscopic characteristics of atmospheric aerosols in Qingdao at specific seasons or pollution levels for radiative transfer modeling, remote sensing, and so forth.

**Keywords:** coastal area; aerosol model; backward trajectory; hygroscopic growth factor





## 1. Introduction

Atmospheric aerosols have complicated impacts on the Earth's climate system via direct and indirect radiative forcing (RF) [1–3]. Behaviors of aerosol optical extinction depend on their microphysical properties (e.g., types, concentration, size distribution) and ambient environmental parameters (e.g., relative humidity, RH; wind speed) [4]. Furthermore, their sources and sinks, and spatial distributions are key uncertainties in current climate models [5]. With the increase of RH, aerosol particles can absorb moisture and undergo a geometric size increase [6], which can significantly enhance the light-scattering and absorbing ability of aerosols [7,8]. The characteristic of the aerosol mentioned above is so-called hygroscopic growth (HG). Accurate simulations of the radiative transfer process require proper considerations of the HG effect [9–13]. However, due to the wide range of temporal and spatial variation of aerosols [14], the over- or under-estimation of aerosol optical properties in describing HG characteristics over a certain area will lead to errors in direct or indirect RF modeling [15,16]. This ambient RH effect is extremely

important for quantifying aerosol's RF, the correction of the measured aerosols by lidar, Multi-Axis Differential Optical Absorption Spectroscopy (MAX-DOAS), as well as for the improvement of satellite retrievals with in situ measurements [3]. Detailed knowledge of RH effects on aerosol scattering is important for the development of aerosol model parameterization, to help to better understand the role of HG on climate change [11], radiation transfer modeling, remote sensing and atmospheric pollution analysis [3,17–19].

Aerosol sources are very complicated in coastal environments, because the coast is a junction area of sea and land. Local aerosols at the coast are jointly affected by various types of aerosols, such as industrial emissions from inland, marine aerosols and mixed aerosols [20,21]. Due to the difference of absorption and scattering characteristics, the climatic and environmental effects of different aerosol types show significant discrepancies [22]. Recently, numerous studies have been carried out on the aerosol radiative transfer and climatic effects in coastal areas. Some have directly observed the weather system and meteorological environment parameters [23,24], while others have used ground-based or satellite-borne remote sensing data to study the spatial and temporal distributions [25,26] and vertical structure of aerosol optical properties [27,28]. There have also been plenty of studies on atmospheric chemistry, which analyze the emissions impact of inorganic [20] and organic [29] black carbon [30] by determining the specific chemical composition of coastal aerosols. However, due to the wide distributions, rapid changes and complex sources of aerosol in coastal areas, a reliable large-scale aerosol model of coastal atmospheric radiation has not been established [3,4,11]. The lack of consideration of intricately mixed aerosols [12,13] and ignoring the critical role of RH in coastal aerosol optical properties [31] will lead to uncertainties in radiative transfer calculation, short-term weather forecasts and climate modeling, the interpretation of the obtained lidar echoes and the performance evaluation of photoelectric detection systems [4,15,16]. Therefore, it is of great significance to acquire aerosol HG characteristics to establish aerosol models that can accurately describe the complex conditions in coastal areas [4].

Efforts on studying aerosol HG could be divided into three types according to the measurement principles: mass HG, particle size HG and optical HG [11,32,33]. The optical HG can be characterized by the aerosol extinction HG factor ($HGF_{ext}$). Compared with the former two (i.e., mass HG and particle size HG [34,35]), the optical HG can directly describe the light-scattering characteristics of environmental aerosols from a macroscopic perspective, which is not limited by the unknown ambient aerosols microphysical parameters (e.g., chemical composition, mixing state, particle morphology), so that the ambient aerosols $HGF_{ext}$ could be directly applied to the calculations of RF [11], which is of great significance to the evaluation of the aerosol climate effect, and has attracted more and more attention.

A great deal of studies have been performed on the coastal $HGF_{ext}$. These include directly measuring the variation characteristics of coastal aerosol $HGF_{ext}$ [36], studying the relationship among pollution degree, aerosol mixing state, chemical composition and hygroscopicity through ground-based observations [37,38], and analyzing the influence of coastal and continental air mass chemical composition on optical hygroscopicity through a spectrometer and a Humidified Tandem Differential Mobility Analyzer (HTDMA) [39,40]. The aerosol's hygroscopicity shows significant dependency on emission sources [41]. When the main source of aerosol is stable, the change of the aerosol number concentration spectrum will have a great influence on the HG characteristics of particle clusters [11]. However, these recent studies only considered the influence of aerosol types on $HGF_{ext}$ retrieval, while another important parameter, the aerosol concentration, was neglected. The aerosol $HGF_{ext}$ inversion algorithm proposed by Liu et al. [23] took the influence of the particle number concentration (PNC) into account, but the $HGF_{ext}$ of different aerosol types was not analyzed in detail due to the short experimental observation period. Therefore, the retrieved $HGF_{ext}$ in the studies above can only represent the aerosol HG characteristics in a short temporal range [35,42,43].

In order to overcome the shortcomings above, we improve the HGF$_{ext}$ retrieval algorithm with a simultaneous inclusion of aerosol type and PNC. The aerosol types are distinguished according to its sources estimated by the backward trajectory model. Aerosols in Qingdao during the filed campaign were classified into oceanic, terrestrial and mixed types, and their characteristics of extinction and HG were then analyzed. This method eliminates the influence of aerosol concentration changes on HGF$_{ext}$ retrieval results and makes HGF$_{ext}$ only dependent on aerosol type.

## 2. Materials and Methods

### 2.1. Data

Qingdao is located on the south coast of Shandong province, China, and is near the boundary of the Bohai Gulf and the Yellow Sea (Figure 1). The meteorological environment on Qingdao is influenced by both interior land and the sea. External aerosols are mostly inland polluted aerosols. Studying the characteristics of aerosol in Qingdao is helpful to understand the aerosols in the coastal areas of China. From 25 September 2019 to 25 October 2020, a long-term field campaign was conducted by the Anhui Institute of Optics and Fine Mechanics (AIOFM), Chinese Academy of Sciences, at the Qingdao Meteorological Station (36.3°N, 120.18°E) for about a whole year. We collected near-surface meteorological and environmental data during the experiments. To better understand the large-scale aerosol properties, we also used officially released weather data, including Global Data Assimilation System (GDAS) data and Moderate-resolution Imaging Spectroradiometer (MODIS) data.

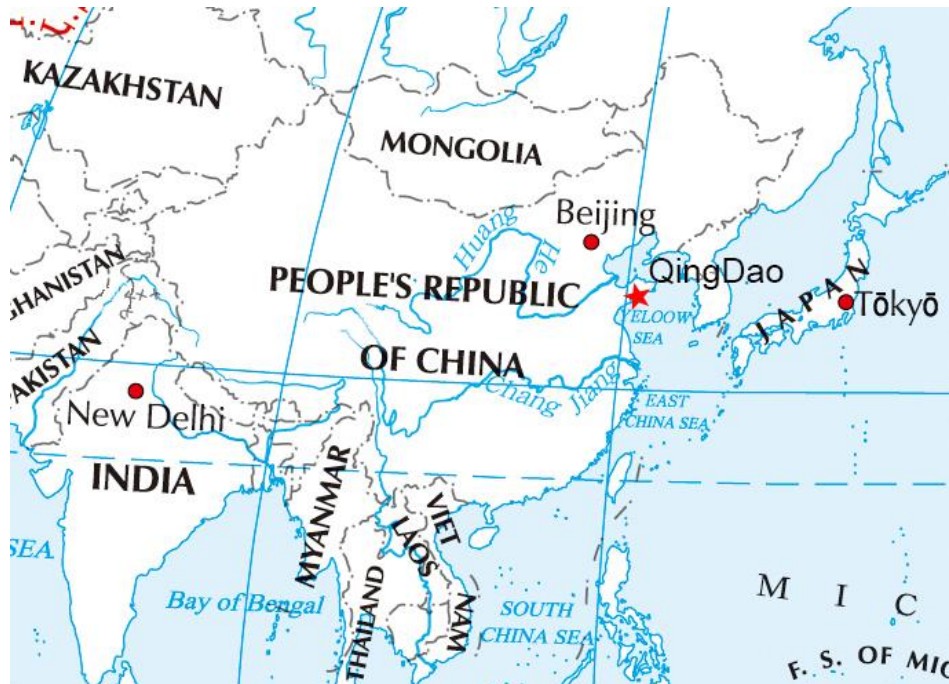

**Figure 1.** Location of Qingdao Meteorological Station (red star), China. The map was provided by the official website of the Ministry of Natural Resources, PRC, NO: GS (2016)2959.

### 2.1.1. Field Experiment Data

We focused on several near-surface meteorological and environmental parameters in this campaign, such as the visibility (VIS) collected by a forward-scattering visibility meter, RH measured by a near-surface automatic weather station, and the particle size distribution (PSD) data acquired by an optical particle counter (OPC). The technical specifications and temporal resolutions of different instruments are shown in Table 1.

1.  The automatic weather station WXT520 (Vaisala, Helsinki, Finland) was used to collect near-surface meteorological data, equipped with 6 weather sensors to measure the atmospheric pressure (P), atmospheric temperature (T), RH, wind direction (WD), wind speed (WS). The WXT520 was mounted on a flux-measuring tower at 2 m above the ground to collect the meteorological data every 5 s during the experiment. The specific performances of the WXT520 sensors refer to Villagrán's paper [44].

2.  The 6000 forward-scattering visibility meter (Belfort Instrument, Baltimore, MA, USA) was used to collect the VIS. The infrared LED transmitter transmits the light into a sampler, and the receiver collects the forward-scattered light and calculates the extinction to obtain the atmospheric VIS, and it is one of the most commonly used instruments for atmospheric visibility measurement. The device can be deployed and used without frequent maintenance. The temporal resolution of the VIS is 5 s. For more technical parameters, please refer to Dr. Dongsheng Ji's research [45].

3.  The OPC was used to collect the aerosol PNC. The OPC is designed to measure aerosol particles such as floating dust. Theoretically, the PNC and the PSD are obtained through the light scattering characteristics of the particles according to the Mie scattering theory [46]. In this paper, we used the DLJ-92 multi-channel OPC developed by the AIOFM, CAS [47]. The detection range of particle diameter size is 0.3–12 μm [47]. The temporal resolution of the OPC data is 60 s in this research.

**Table 1.** Technical specifications and temporal resolutions of measurement instruments.

| Technical Specification | Data | Measurement Range | Accuracy | Resolution | Temporal Resolution |
|---|---|---|---|---|---|
| WXT520 | RH | 0–100% RH | ±3% (0–90%RH) ±5% (90–100%RH) | 0.1% RH | 5 s |
| Belfort 6000 | VIS | 200 m~50 km | ±10% | \ | 5 s |
| OPC | PNC | 0.3~12 um | \ | 17–20 channels optional | 60–600 s optional |

### 2.1.2. Remote Sensing and Reanalysis Data

We also used officially released GDAS and MODIS aerosol optical depth (AOD) data in our analysis. The GDAS was utilized in the backward trajectory simulations to analyze the aerosol types, and the MODIS AOD data were used to analyze the aerosol pollution during the experiment.

(1)  GDAS data

We used the GDAS data as the driving data for HYSPLIT, a flow-backward tracking simulation software, to analyze the flow-backward tracking and the aerosol source regions during the Qingdao field campaign. GDAS data can be downloaded via ftp://arlftp.arlhq.noaa.gov/pub/archives/ (accessed on 28 January 2021). This dataset is capable of estimating and predicting the global atmospheric state every six hours. The GDAS data assimilates ground-based observations, radiosonde data, aircraft spot detections, and shipboard and satellite observations [48], which are used to obtain downward radiation, T, RH, WS, P, and precipitation data near the surface. In this paper, we collected the global GDAS data from July 2019 to August 2020. The temporal resolution is 1 h, and spatial resolution is $0.25° \times 0.25°$.

(2)  MODIS AOD data

In addition, we used AOD data from MODIS/Aqua to estimate the air pollution level from aerosol sources. MODIS AOD data is a widely used dataset in weather forecasting, detection and assessment, which can help to better understand the characteristics of global climate change [49]. A large number of studies have shown that the AOD results obtained by MODIS inversion are highly consistent with those obtained by ground-based remote sensing inversion (Data download address: https://ladsweb.modaps.eosdis.nasa.gov/search/order/1/MODIS: Aqua (accessed on 7 August 2020)) [49]. We used the MODIS

Level-3 product data MYD08M3 to derive local aerosol optical properties. The data set of MYDO8M3 is AOD_550_Dark_Target_Deep_Blue_Combined_Mean_Mean, obtained by the dark target and deep blue algorithm [50,51]. The temporal resolution is 1 month, and spatial resolution is 1 km.

### 2.2. Methodology

In order to improve the accuracy of HGF$_{ext}$ retrieval and to provide a theoretical basis for the study of aerosol optical properties in coastal regions, the effects of different aerosol sources and aerosol concentrations should be taken into account. The backward trajectory tracing of local aerosol is used in our work, and the HGF$_{ext}$ inversion algorithm was proposed by Liu et al. [23] which takes the PNC as input to exclude the influence of aerosol concentration changes on the HGF$_{ext}$ inversion results. The theoretical basis of the proposed algorithm is the relationship between VIS, extinction, and hygroscopicity. Liu et al. [23] have specified the relationships between VIS and PNC, RH, and WS, in the coastal area. The results show that WS exhibit weak relationships with VIS, but PNC and RH both exhibit negative correlations with VIS, and with the increasing of RH, VIS exhibited a sharper exponential decline. According to the Mie scattering theory, VIS can be expressed as a function of PNC and $\delta_{ext}$ (aerosol particle clusters' mean extinction coefficient). Therefore, the mean extinction coefficient can be expressed as Equation (1).

$$\delta_{ext} = \frac{3.912}{VIS \times PNC}, \tag{1}$$

HGF$_{ext}$ can be obtained from the dry–wet ratio of the mean extinction coefficient (Equation (2)) [52].

$$HGF_{ext} = \frac{\delta_{ext,wet}}{\delta_{ext,dry}}, \tag{2}$$

the $\delta_{ext,dry} = \delta_{ext}(RH = 0)$ [53], but there is no completely dry aerosol (RH = 0%) in the field campaign, so the aerosol in a relatively dry environment is generally considered as the dry aerosol [54]. In this research, we use $\delta_{ext}(RH < 50\%)$ for $\delta_{ext,dry}$, and $\delta_{ext}(RH > 50\%)$ for $\delta_{ext,wet}$.

Observations in coastal area of Guangdong province in China showed that the observed negative correlation between VIS and PNC in each RH bin is consistent with Equation (1) [23]. Therefore, it is practicable to study aerosol HGF$_{ext}$ with this method. In this paper, we combined this method with an aerosol type classified by backward trajectory tracing to study the different aerosol types' HG characteristics by utilizing long-term observations in Qingdao.

## 3. Results and Discussion

### 3.1. Monthly and Seasonal Characteristics of Atmospheric Aerosols in Qingdao

According to (Equations (1) and (2)), VIS is an important input parameter, and the accuracy of VIS measurements will directly affect the retrieved HGF$_{ext}$. Therefore, a preprocess of the raw VIS observations is needed to remove the spines and the unreasonable points, and thus to avoid introducing large errors.

### 3.1.1. Data Preprocessing

We evaluated the performance of three burr point-removing methods in the raw VIS data, including the 10-point moving average method, Butterworth low-pass filtering, and a delay-free Butterworth low-pass filtering method. For the moving average method, we selected a window of 10 points on original data and obtained the mean value of all data by adding the new data and subtracting the old data sequentially to eliminate accidental variability factors, at the same time. For the filtering method, we constructed a Butterworth low-pass filter to remove the noise signal (burr points) in original VIS data, which uses the characteristic that the noise signal frequency is higher than the original signal. Since it

involves the conversion of signals from the time domain to the frequency domain, the use of filters will result in a delay of the filtered signals. To address this issue, we processed the raw VIS data using the delay-free Butterworth low-pass filter. The basic parameters of these two filters are the same: cut-off frequency, 0.8; sampling frequency, 32; order, 4. The results of VIS data preprocessing are shown in Figure 2, where the black line represents the raw data, the green one represents the 10-point moving average result, the blue one represents the Butterworth low-pass filtering result, and the red one represents the non-delayed Butterworth low-pass filtering result.

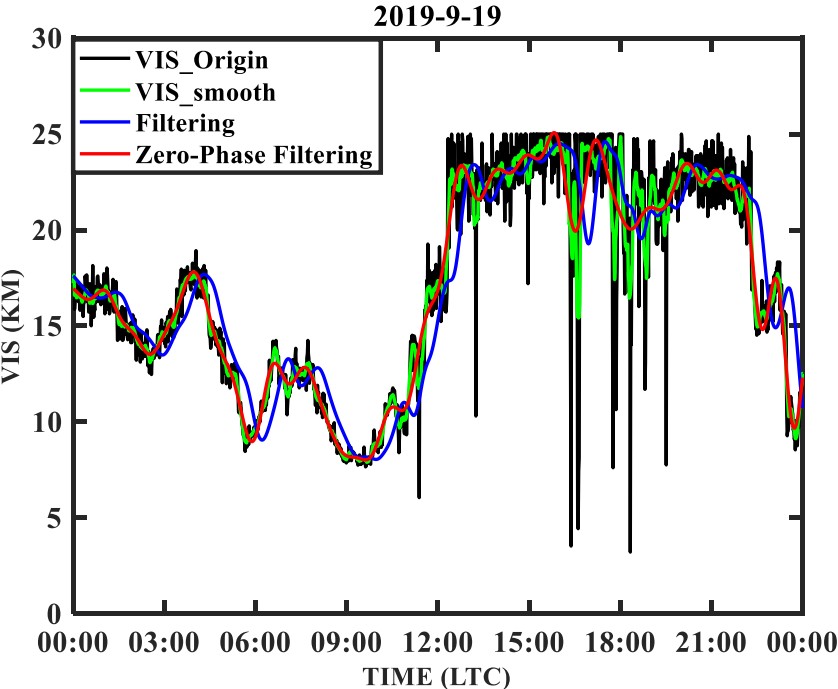

**Figure 2.** VIS preprocessing results, including: raw data (black line); 10-point moving average (green line); Butterworth low-pass filtering (blue line), and non-delay Butterworth low-pass filtering (red line).

As shown in Figure 2, the results from different data preprocessing methods are quite similar, but there are still some differences. The 10-point moving average result (green line) is more consistent with the raw signal, but at the locations where noise signals continuously appear (such as 3:00 p.m. to 6:00 p.m. on 19 September 2019), it resulted in errors that cannot be ignored. In the data preprocessing by the Butterworth low-pass filter method, the raw signal needed to be transformed and processed from the time domain to the frequency domain.

The filtered signal (blue line) has a time domain delay compared with the raw VIS signal (black line). However, it can be seen that the noise in the original signal can be removed by using the filter method. Compared with VIS results using the former two methods, the preprocessed VIS (red line) by use of the non-delay Butterworth low-pass filter is much better, which can eliminate large errors from the moving average method and the delays that occur due to time-frequency transformation of the Butterworth low-pass filtering method. Therefore, the raw VIS data were preprocessed using the delay-free Butterworth low-pass filter in this work, for the consistency trend between the final results and the original data.

From data preprocessing results, we can see the diurnal asymmetry of VIS data in this study, as shown in Figure 2. The diurnal feature of VIS was found to be at maximum from 12:00 p.m. to 10:00 p.m. LTC, and after that it started falling. This diurnal asymmetry pattern can also be found in other months. This diurnal asymmetry of VIS is consistent with

the research results of recent studies near Qingdao (in Huludao [55] and Jiangsu province, China [56]).

### 3.1.2. AOD and VIS in Qingdao

We used the VIS data measured by the 6220 forward-scattering VIS meter and the monthly AOD of 550nm from the MODIS/Aqua level 3 standardized data product MYD08M3 to analyze the monthly and seasonal aerosol characteristics in Qingdao. The monthly MODIS AOD distributions over Qingdao and adjacent areas, from September 2019 to August 2020, are shown in Figure 3. The four seasons in following discussions are de-fined as: spring (March to May 2020, Figure 3(c1–c3)), summer (June to August 2020, Figure 3(d1–d3)), autumn (September to November 2019, Figure 3(a1–a3)), and winter (December 2019 to February 2020, Figure 3(b1–b3)).

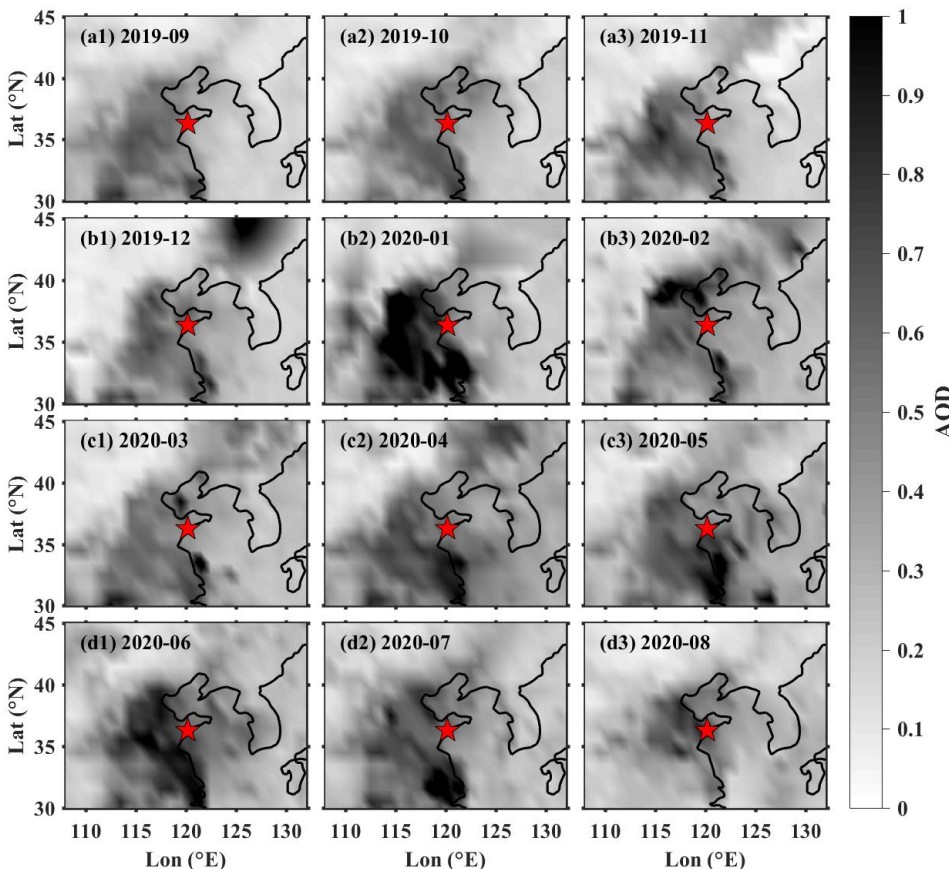

**Figure 3.** Monthly MODIS AOD from September 2019 to August 2020. The red five-pointed star in each plot denotes the location of Qingdao.

From the monthly AOD in Figure 3, we can see that autumn has the smallest AOD among four seasons in Qingdao. It is worth mentioning that a large region near Qingdao was in a heavy pollution situation in January 2020, including the Hebei, Shandong, Henan, Anhui, and Jiangsu provinces. One potential reason is that the urban heating increases the air pollution in winter. However, the AOD in this region has shown a downward trend since February 2020. One possible reason ought to be that China began blocking the city from January 2020 and stopped the production of industrial enterprises during this period because of the new coronavirus disease (COVID-19), resulting in weakened air pollution in a large area. Small-scale pollution incidents are concentrated in densely populated cities, such as Beijing and Shanghai, because of the large scale of heating in winter for citizens in these cities. When the epidemic was effectively contained, citizens resumed work and industrial pollution production gradually increased from April 2020.

The variations of VIS during the observations period are shown in Figure 4. The blue solid line represents the raw observed VIS, and the red dotted line represents the low-pass filtered VIS.

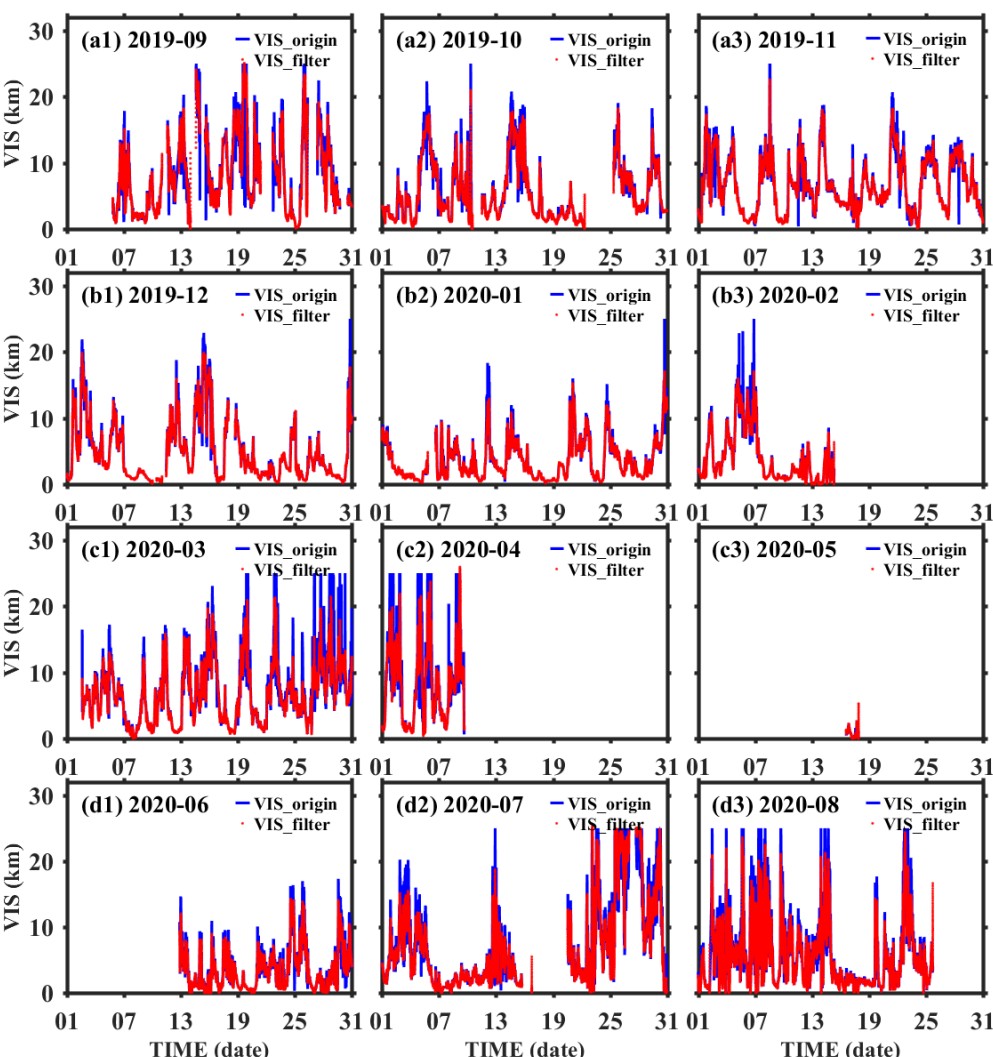

**Figure 4.** Variations of VIS in Qingdao from September 2019 to August 2020, including raw data (blue solid line) and low-pass filtered data (red dotted line).

Previous studies showed that AOD have diurnal asymmetry along with the VIS data [57,58], which can also be found in our results, as shown in Section 3.1.1. At the same time, AOD is negatively correlated with VIS [59], but this relationship is not a simple linear relationship, and is different in each season [60]. Thus, seasons and meteorological conditions need to be taken into consideration in studies. In this research, the AOD also negatively correlated to VIS, as showed in the monthly AOD spatial distributions (Figure 3) and monthly VIS in Qingdao (Figure 4). In autumn with relatively lower AOD in Qingdao, higher VIS values than other seasons (i.e., spring, summer, and winter) can be observed. When the pollution levels were high in January 2020, as suggested by AOD, the VIS values were as low as 15 km or less. Note that there is a lack of VIS observational data during 2020–05, because the field campaign was interrupted in this month due to COVID-19.

### 3.2. Analysis of Aerosol Source

The HYbrid Single-Particle Lagrangian Integrated Trajectory (HYSPLIT4) model, implemented in the meteorological analysis mapping software MeteoInfo with the TrajStat plug-in, was used to analyze the aerosol sources in Qingdao. The HYSPLIT4 model, jointly

developed by the US Air Resources Laboratory and the Australia Bureau of Meteorology, was employed for the analysis of the transport and diffusion trajectories of pollutants and for simulating the backward trajectories of air masses [61]. TrajStat displayed the air mass trajectories [62,63].

In the simulation, the Institute of Marine Instruments, Shandong Academy of Sciences Station (36.3°N, 120.18°E) was selected as the starting point with the simulated height of 1500 m above the surface. The GDAS database was used as meteorological inputs for HYSPLIT4 to simulate the backward trajectories with durations of 48 h at a time resolution of 1 h. The aerosol backward trajectories' cluster results are shown in Figure 5, overlayed with monthly mean MODIS AOD.

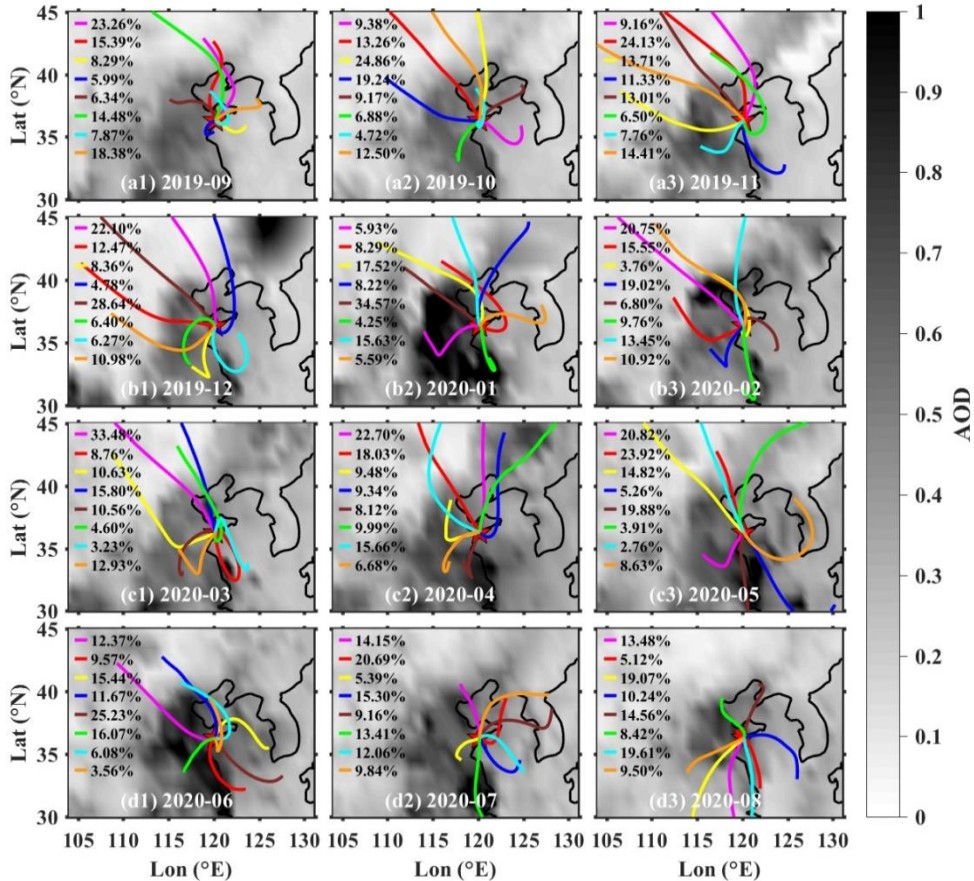

**Figure 5.** Cluster results from September 2019 to August 2020 in Qingdao.

We used a two-step classification method to statistically classify aerosol types. First, the backward trajectories of the Qingdao airmass were grouped into eight clusters using MeteoInfo software, as shown in Figure 5. In the legends, the backward tracks of air mass with eight different colors from top to bottom are marked as Numbers 1–8 air mass sources (i.e., purple, red, yellow, blue, black, green, light blue and orange). Second, eight clusters were further classified into three aerosol types (pure marine, pure terrestrial, mixed sources) according to air mass originations, to simplify the recognition of the coastal areas' aerosol types. In the classification, aerosol originated from the Yellow Sea is considered as clean marine aerosol. Aerosols from the Bohai Sea are identified as mixed-source aerosol, with marine aerosols mixed with those from the cities in northeast China, Inner Mongolia, Hebei and other provinces. Aerosols from inland are polluted urban aerosols (pure terrestrial). Finally, we could simply revise the classification results according to the aerosol extinction coefficients of the three aerosol types, because this classification method may result in misclassification of some clusters into pure terrestrial or marine sources which actually belong to the source-mixed type. This phenomenon can also be found in subsequent results,

which will be discussed in Section 3.3, and in this section we only discuss the results of two-step classification. The aerosol type classification indexes based on the clustering results are shown in Table 2, and the percentages of each class based on the classification indexes are shown in Table 3 and Figure 6.

**Table 2.** Aerosol source index classification results from backward trajectory clustering.

| Time | Land Source Index | Sea Source Index | Mixed Aerosol Index |
|---|---|---|---|
| September 2019 | 5 | 3, 8 | 1, 2, 4, 6, 7 |
| October 2019 | 2, 4, 6 | 1 | 3, 5, 7, 8 |
| November 2019 | 3, 5, 7, 8 | 6 | 1,2,4, |
| December 2019 | 2, 3, 5, 6, 8 | 7 | 1, 4 |
| January 2020 | 1, 5 | 8 | 2, 3, 4, 7, 8 |
| February 2020 | 1, 2, 4, 6 | 5 | 3, 7, 8 |
| March 2020 | 1, 3, 5, 8 | 2, 7 | 4, 6 |
| April 2020 | 2, 3, 5, 7, 8 | 4 | 1, 6 |
| May 2020 | 1, 3, 4 | 5, 8 | 2, 6, 7 |
| June 2020 | 1, 6 | 2, 5 | 3, 4, 7, 8 |
| July 2020 | 3, 6 | 4, 7 | 1, 2, 5, 8 |
| August 2020 | 1, 3, 6, 8 | 2, 4 | 5, 7 |

1—purple, 2—red, 3—yellow, 4—blue, 5—black, 6—green, 7—light blue and 8—orange.

**Table 3.** Aerosol source percentage results from clustering of backward trajectories.

| Time | Land Source (%) | Sea Source (%) | Mixed Aerosol (%) |
|---|---|---|---|
| September 2019 | 6.34 | 26.67 | 66.99 |
| October 2019 | 39.38 | 9.38 | 51.25 |
| November 2019 | 48.88 | 6.5 | 44.62 |
| December 2019 | 66.85 | 6.27 | 26.88 |
| January 2020 | 40.5 | 5.59 | 55.25 |
| February 2020 | 65.08 | 6.8 | 28.13 |
| March 2020 | 67.6 | 11.99 | 20.4 |
| April 2020 | 57.97 | 9.34 | 32.69 |
| May 2020 | 40.9 | 28.51 | 30.59 |
| June 2020 | 28.44 | 34.8 | 36.75 |
| July 2020 | 18.8 | 27.36 | 53.84 |
| August 2020 | 50.47 | 15.36 | 34.17 |

It can be seen from the backward trajectory cluster classifications (Table 2) that the aerosol source in Qingdao is complex. Most aerosols are from land sources (northwest of Shandong province) and the mixed aerosols (northeast of Shandong province) are also superior in numbers, and only a small number of aerosols came from sea sources (southeast of Shandong province from the Yellow Sea).

As shown in Figure 6, there are significant monthly variations in the fraction of backward trajectories from different sources. The proportion of land sources increased from ~6% to ~48% in autumn, and reached the maximum (~65%) in winter and spring, then it gradually decreased to ~18% in summer. In contrast to the land-source aerosol, there is an opposite trend in the fraction of the aerosols from sea and mixed sources. The sea sources' proportion decreased from ~26% to ~6% in autumn and reached the minimum (~5.5%) in winter and spring, then increased gradually to ~34% in summer. The proportion of mixed source aerosol is more than that of the marine source one, but the monthly variation is close to that of marine aerosol. Therefore, it is necessary to distinguish aerosol sources in coastal aerosol optical properties research, because the fraction of different aerosol sources proportion varies so significantly. In the next section, we will apply these statistical results in the analysis of HGF$_{ext}$ characteristics with different aerosol sources.

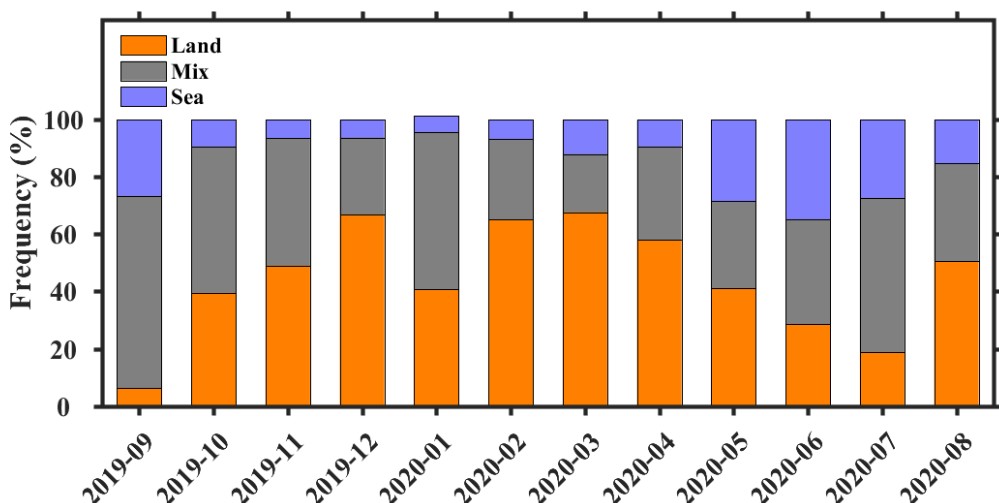

**Figure 6.** Percentage of different aerosol sources from September 2019 to August 2020 in Qingdao based on backward trajectory clustering.

### 3.3. Characteristics of Aerosol Extinction HG Factor

### 3.3.1. Monthly HGF$_{ext}$ Characteristics in Qingdao

We used the algorithm mentioned in Section 2.2 to retrieve the aerosol HGF$_{ext}$ from different sources in Qingdao. The data processing steps are as follows:

(1) Select the time of all backward trajectories in each clustering result;
(2) Match the VIS, PNC and RH values according to the selected time in Step 1;
(3) Calculate the average extinction coefficient ($\delta_{ext}$) of each clustering result using Equation (1) to analyze the relationship between $\delta_{ext}$ and RH, then form a new array against RH;
(4) Classify all the $\delta_{ext\_land}$, $\delta_{ext\_sea}$, $\delta_{ext\_mix}$ corresponding to different aerosol sources in each month, using the aerosol source results identified in Table 2;
(5) Calculate the aerosol extinction HG factor (HGF$_{ext\_land}$, HGF$_{ext\_sea}$ and HGF$_{ext\_mix}$) for all aerosol types using Equation (2), and establish the power-law curve models for different aerosol types.

The monthly $\delta_{ext}$ distributions of each cluster during the experimental period are shown in Figure 7. The monthly $\delta_{ext}$ variation trends for all eight aerosol clusters show only minor discrepancies with each other. However, some clusters have the same trend of variation due to similar aerosol properties, which illustrates the necessity of statistically classifying the eight clusters into three categories.

Then, we used the statistical results in Table 2 to classify the different aerosol sources. We used Equation (2) to calculate the HGF$_{ext\_land}$, HGF$_{ext\_sea}$ and HGF$_{ext\_mix}$ of each month, and the nonlinear fitted results are shown in Figure 8. The single-parameter equation proposed by Kasten [64] was used in the fitting, where *A* and *B* are the fitting coefficients.

$$y = A(1 - \text{RH}/100)^{-B}, \tag{3}$$

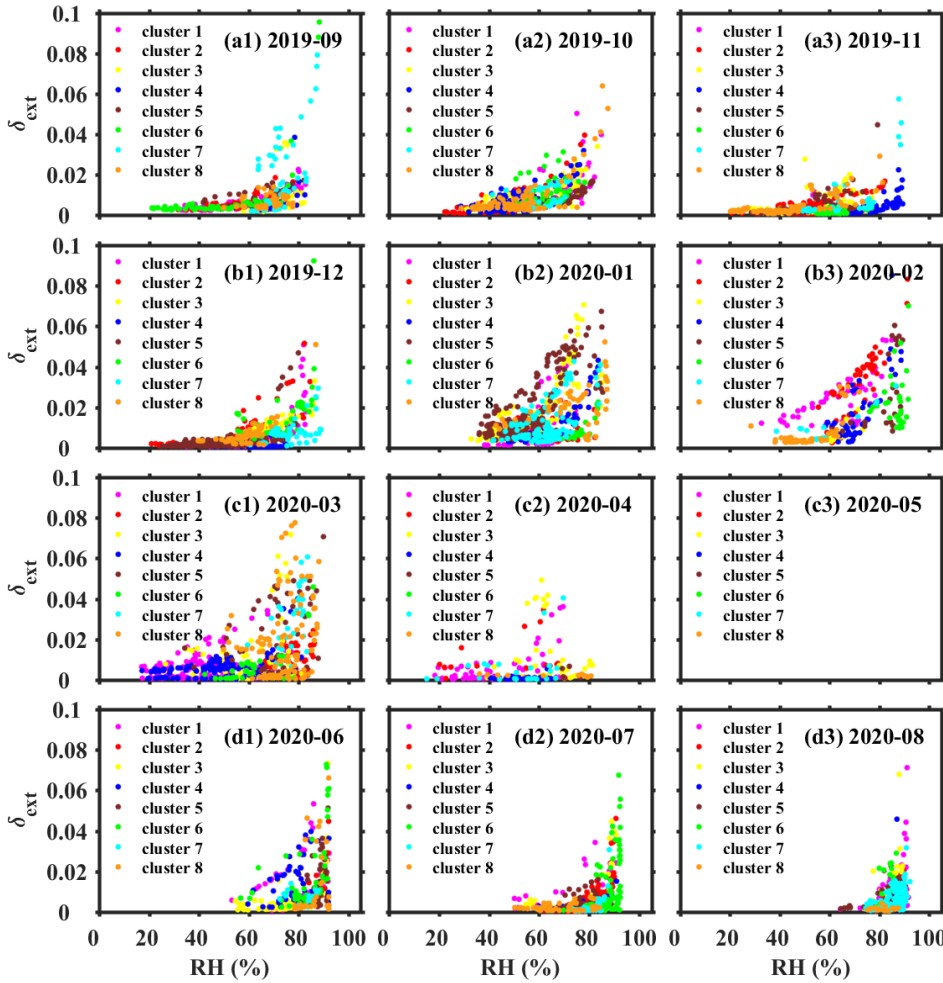

**Figure 7.** Monthly $\delta_{ext}$ distributions of each kind of backward trajectories' clusters in Qingdao from September 2019 to August 2020.

Figure 8 shows that the local aerosols' $HGF_{ext}$ from different sources in Qingdao have relatively small monthly variations, especially in months with insufficient observational data, and the fitting results were apparently wrong, such as the land source $HGF_{ext}$ in April 2020 (Figure 8(c2)), the red line, due to the lack of data in the range of RH > 80%). However, we could find that the $HGF_{ext}$ features are very similar in the months belonging to the same season, such as shown in Figure 8(a1–a3). At the same time, the $HGF_{exts}$ features vary considerably between seasons. For example, the $HGF_{exts}$ deliquescence point (DP) in autumn is at 50 < RH < 80% (September to November 2019, Figure 8(a1–a3)), but in summer it is around RH = 90% (June to August 2020, Figure 8(d1–d3)).

In addition, in Figure 8(b1,c1), it can be observed that the land-originated aerosol data points (red dots) are distributed more discretely, and the fitted curve (red line) seems to split the data points into two independent parts (above and below the fitted curve). The distribution trend of data points below the fitted curve of terrestrial source (red curve) in Figure 8(b1) is closer to the fitted curve of the mixed-source one (blue curve). It can be found in Figure 7(b1) that the data points in this position belong to cluster 5, and Figure 5(b1) shows that the trajectory of this cluster originates from inland but passes through the Bohai coast. Although it is classified as the land source aerosol, it exhibits the characteristics of source-mixed aerosol to a certain extent, indicating that the physical properties of the air mass are affected by marine aerosol. The situation in Figure 8(c1) is similar to that in Figure 8(b1); if the data below the fitted land source curve is separately normalized, its characteristics are closer to that of the mixed type. Therefore, it is necessary to revise some aerosol classification results in Section 4 according to the specific characteristics of $HGF_{ext}$.

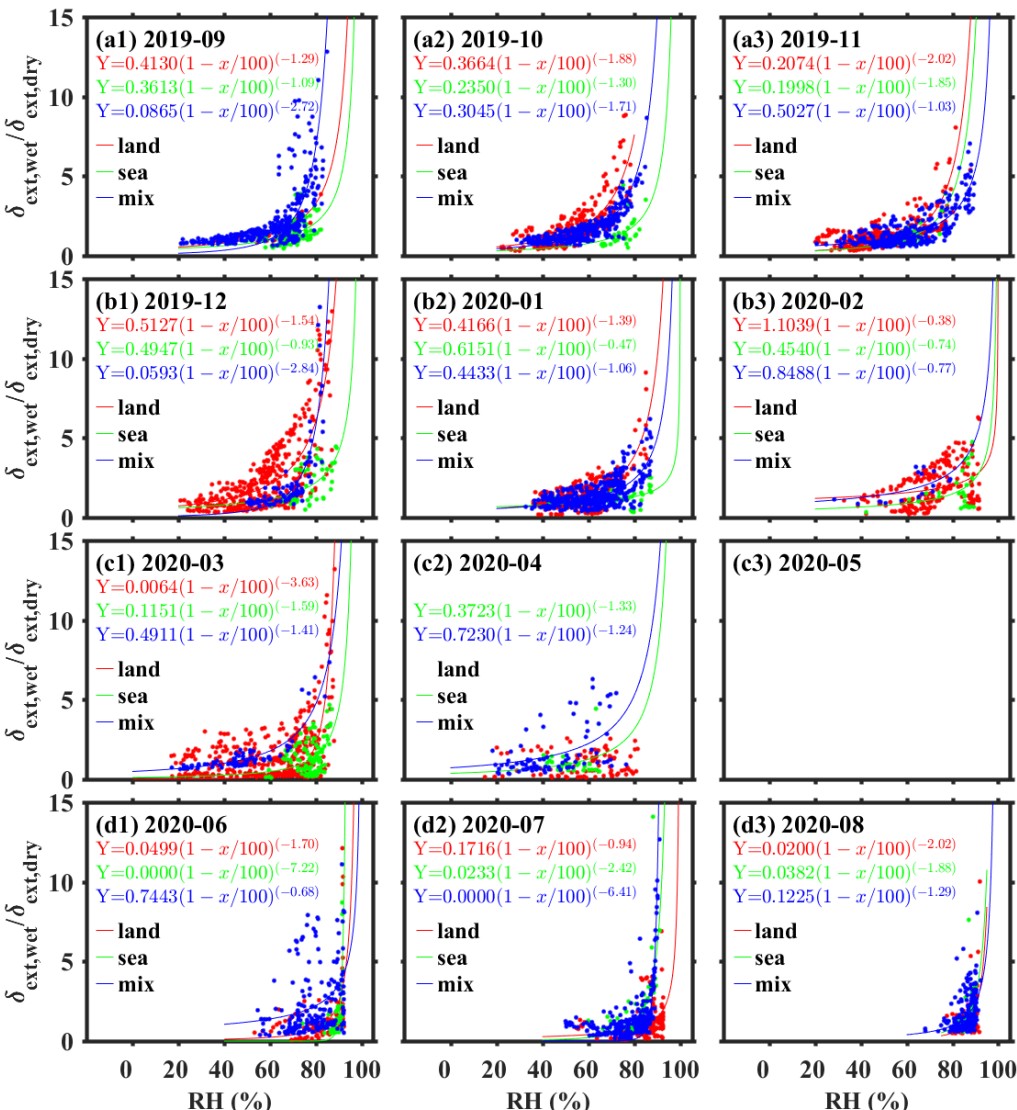

**Figure 8.** Monthly HGF$_{ext}$ of aerosol from different source in Qingdao from September 2019 to August 2020.

In the next section, we simply revise the monthly aerosol classification, taking the seasonal variations into account, and obtain the seasonal HGF$_{ext\_land}$, HGF$_{ext\_sea}$ and HGF$_{ext\_mix}$ regression models for Qingdao according to the polynomial fitting method.

### 3.3.2. Seasonal HGF$_{ext}$ Characteristics in Qingdao

In this section, we investigate the HGF$_{exts}$ seasonal variation characteristics, and use numerical simulation to obtain the HGF$_{exts}$ regression model for all seasons in Qingdao (Figure 9). The regression model evaluation indexes include: RMSE and R-square. The regression results are shown in Figure 9. The fitting parameters and evaluation indexes (RMSE, R-square) of the fitting results are shown in Table 4.

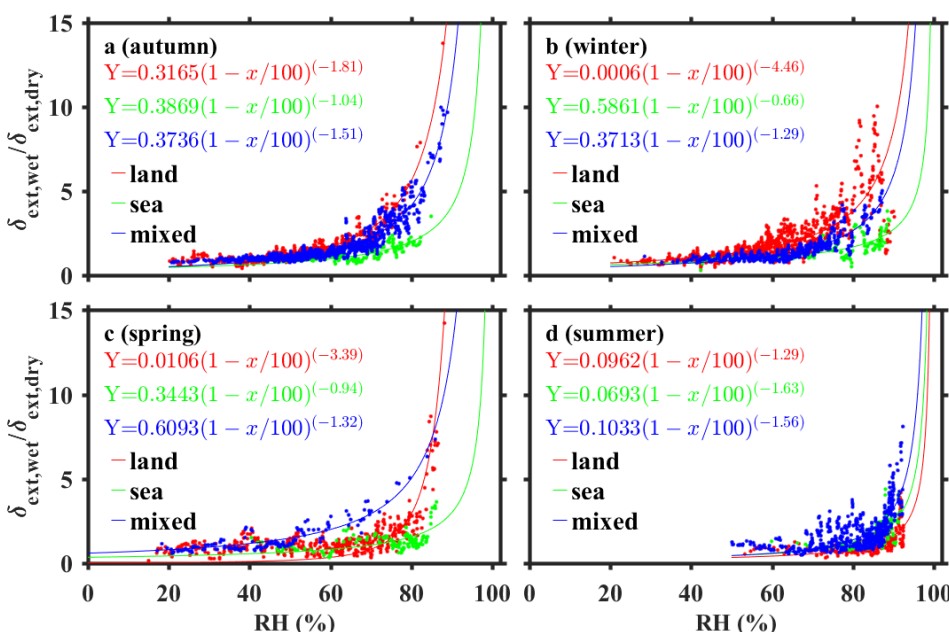

**Figure 9.** Seasonal HGF$_{ext}$ of different sources of aerosol in Qingdao from September 2019 to August 2020: (**a**) stands for autumn, (**b**) for winter, (**c**) for spring, (**d**) for summer.

**Table 4.** Fitting parameters of HGF$_{ext}$ and evaluation indexes for each season model.

| Season | Land Source | | | | Sea Source | | | | Mixed Aerosol | | | |
|--------|------|------|------|----------|------|------|------|----------|------|------|------|----------|
|        | A | B | RMSE | R-Square | A | B | RMSE | R-Square | A | B | RMSE | R-Square |
| autumn | 0.3165 | 1.81 | 0.34 | 0.92 | 0.3869 | 1.04 | 0.26 | 0.65 | 0.3736 | 1.51 | 0.42 | 0.89 |
| winter | 0.0006 | 4.46 | 1.52 | 0.46 | 0.5864 | 0.66 | 0.47 | 0.49 | 0.3713 | 1.29 | 0.35 | 0.79 |
| spring | 0.0106 | 3.39 | 0.89 | 0.89 | 0.3443 | 0.94 | 0.54 | 0.32 | 0.6093 | 1.32 | 0.35 | 0.89 |
| summer | 0.0962 | 1.29 | 0.52 | 0.41 | 0.0693 | 1.63 | 0.51 | 0.50 | 0.1033 | 1.56 | 0.78 | 0.49 |

The aerosol HG characteristics can be evaluated using DPs, and the aerosol HGF$_{ext}$ increases rapidly when RH is higher than the DP, and a smaller DP value implies a stronger hygroscopicity. Conversely, higher DP implies weaker hygroscopicity. It can be seen from the fitting results (Figure 9) that the HGF$_{ext\_seas}$ for marine aerosol (in four seasons) is very similar in different seasons, as the DPs were all around RH = 80% and the HGF$_{ext\_seas}$ increased rapidly when RH is greater than 80. However, the HGF$_{ext\_land}$ and HGF$_{ext\_mix}$ for land with original aerosol and source-mixed aerosol, respectively, were much more complicated over the four seasons.

In spring, the aerosols' hygroscopic properties (Figure 9c) were more complex compared with the other seasons, especially for the HGF$_{ext\_mix}$ of which the DP was as low as RH = 50%. In the case where the RH was lower than 85%, the HGF$_{ext\_mix}$ was greater than HGF$_{ext\_land}$, and when RH increased to more than 85%, the HGF$_{ext\_land}$ was greater than the source-mixed one (HGF$_{ext\_mix}$).

In summer, all three aerosol types' HGF$_{ext}$ characteristics were similar, and are quite similar to the HGF$_{ext\_seas}$ with DPs of about 80%. This indicates that the atmospheric circulation in Qingdao is relatively strong in summer, so that the aerosol mixed well, and the aerosol properties of all three sources are consistent. The reason may be that Qingdao's local atmosphere in summer is greatly affected by the sea breezes from the southeast, and the large-scale weather systems bring clean ocean-derived aerosols deep into the land, so Qingdao is under an atmospheric background filled with marine aerosols. Even a small-scale circulation blows land breeze, and it carries the sea-derived aerosols trapped in the inland.

In autumn and winter (Figure 9a,b), the DPs of the terrigenous and mixed sources aerosol are about RH = 60% and 70%, respectively. This difference in DPs suggests that terrestrial aerosols transported inland during autumn and winter have stronger hygroscopicity, while aerosols transported from the sea have poorer optical hygroscopicity. Therefore, it can be inferred that the air pollution at Qingdao in autumn and winter is mainly caused by terrestrial aerosols with stronger hygroscopic properties.

### 3.3.3. HGF$_{ext}$ Characteristics under Different Pollution Backgrounds in Qingdao

In this section, we further compare the HGF$_{ext\_land}$, HGF$_{ext\_sea}$ and HGF$_{ext\_mix}$ according to air pollution levels during the experiment. As shown in Section 3.1.2, Qingdao's AOD obviously decreased and the VIS increased since February 2020, which means that the air quality significantly improved. Such a dramatic change could have been caused by the reduction in emissions and industrial pollution for the nationwide lock-down policy implemented by the Chinese government because of the COVID-19 pandemic during the Qingdao experiment.

Therefore, we divided the observed data in Qingdao into two periods, the relatively polluted period before the outbreak of COVID-19 (period 1, September 2019 to January 2020) and the clean period after the lockdown policy (period 2, February 2020 to July 2020). Then, we performed a nonlinear fit, as shown in Figure 10. Regression models obtained for different sources' aerosol in Qingdao under different pollution backgrounds are shown in Table 5.

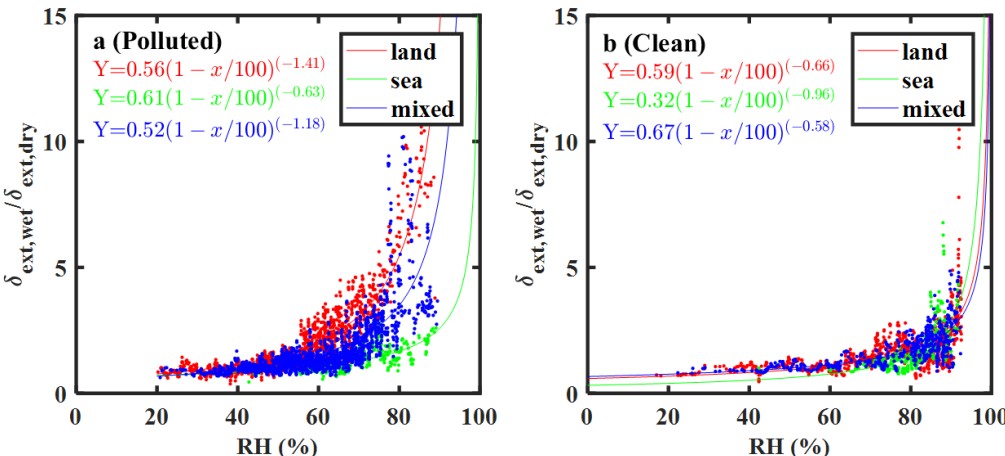

**Figure 10.** Aerosol from different sources under polluted and clean backgrounds in Qingdao from September 2019 to August 2020.

**Table 5.** Fitting parameters of HGF$_{ext}$ and evaluation indexes for different-sourced aerosols in each season.

| Aerosol Source | Period 1 (Polluted) | Period 2 (Clean) |
|---|---|---|
| Land source | y = 0.56(1 − x/100) − 1.41 | y = 0.59(1 − x/100) − 0.66 |
| Sea source | y = 0.61(1 − x/100) − 0.63 | y = 0.32(1 − x/100) − 0.96 |
| Mixed aerosol | y = 0.52(1 − x/100) − 1.18 | y = 0.67(1 − x/100) − 0.58 |

Figure 10 shows that the differences between HGF$_{ext\_land}$, HGF$_{ext\_sea}$ and HGF$_{ext\_mix}$ are more significant in Period 1 than that in Period 2. In Period 1 (Figure 10a), the DPs of the land source, mixed source, and sea source aerosols' HGF$_{ext}$ models are 50% (RH), 60% (RH), and 70% (RH) respectively. This means that the properties of aerosol from different external sources are quite different in Qingdao during this period, and the mixing of the aerosols from external sources with local polluted emissions can strongly change the macroscopic characteristics. During Period 2 (Figure 10b), the features of the aerosols from different external sources are quite similar, for the HGF$_{ext\_land}$, HGF$_{ext\_sea}$ and HGF$_{ext\_mix}$ DPs are

all around RH = 70%. A possible reason could be that the atmosphere was very clean at that time, and the mixing of aerosols from external sources with minimal local emissions had little effect on the optical aerosol properties. The differences between $HGF_{ext\_land}$, $HGF_{ext\_sea}$, and $HGF_{ext\_mix}$ in the clear background are not obvious due to the same reason stated above.

We also find that the regression model of $HGF_{ext\_sea}$ in Period 1 is similar to that in Period 2, implying the aerosol in Qingdao are mainly the rather clear marine aerosols because the atmosphere background is relatively clean. The $HGF_{ext\_mix}$ under the polluted background is close to the observation result of $HGF_{ext\_mix}$ under the polluted background in Maoming in the previous research [23], for the $HGF_{ext\_mix}$ DPs are around RH = 60%.

In addition, previous studies generally agree that the land-source aerosol HG are weaker than that of the sea source aerosol. However, in this study, we found a different phenomenon. The $HGF_{ext\_land}$ DP is RH ~ = 60%, which is lower than that of the $HGF_{ext\_sea}$ (RH ~ = 80%), which means that the HG of the land-source aerosol is stronger than that of the sea source in the heavily polluted background. This feature may be responsible for the frequent occurrence of heavy pollution events. However, the specific cause needs to be investigated further in the future using aerosol sampling and laboratory analysis.

## 4. Conclusions

In this study, we analyzed the monthly and seasonal characteristics of atmospheric aerosols and calculated the aerosol source fraction of pure terrestrial, pure marine and mixed-source aerosol in Qingdao from September 2019 to August 2020. The regression models of $HGF_{ext\_land}$, $HGF_{ext\_sea}$ and $HGF_{ext\_mix}$ in each season and under different pollution backgrounds were obtained, respectively, using the HG-retrieving algorithm in previous research [23] and the aerosol type identification classified by its sources. The main conclusions of this paper are as follows.

- Qingdao's aerosol source is very complicated due to the special geographical location, a typical coastal area in northeastern China. The proportion of different aerosol sources in Qingdao varies greatly from month to month, and shows obvious seasonal fluctuations. The local aerosol sources are mostly terrestrial sources, with marine sources accounting for only about 10–20%, which is slightly higher in summer than that in autumn and winter. Terrestrial aerosols accounted for more than 40% of the total for the year.
- The local aerosol's $HGF_{ext}$ of different sources in Qingdao have relatively small monthly variations, but explicit a certain seasonal variation. The seasonal change is mainly manifested as the "floating" of the DPs, and the DP of marine aerosol (RH about 80%) is consistent in different seasons. The seasonal distributions of terrestrial and mixed aerosols' DPs are different, decreasing as low as to RH = 60–70% in autumn and winter and rising to about RH = 85% in summer. The DPs of mixed- source aerosols are generally intermediate between terrestrial and marine source ones. These variations can be caused by atmospheric circulation and local pollution levels.
- Under the background of different pollution levels, the characteristics of local aerosol from different sources in Qingdao showed considerable discrepancy. In general, when the atmospheric background was relatively clean, the DPs of aerosols from different sources were almost the same (about RH = 80%), but when the pollution was heavy, the DPs of terrestrial aerosols were almost 20% lower than those of marine sources (in the period of heavy pollution, the DPs' RH of terrestrial, mixed and marine aerosols are 50%, 60%, and 70%, respectively). Therefore, it is necessary to model the local aerosols in Qingdao based on different pollution backgrounds.

From the analytical results above, a quantitative $HGF_{ext}$ model of aerosol from different sources in Qingdao under different seasons and pollution background conditions was established, which can be used in the calculation of the aerosol radiation transmission effect in coastal areas, inversion and correction of satellite remote sensing products, and other relevant fields. It is worth mentioning that aerosol type classification could be further

verified and improved by combining other measurements, such as utilizing the relationship between the AOD and Angstrom Exponent (AE) based on spectral AOD measurements. However, the temporal resolution of MODIS data is very coarse (only one record per day for each satellite), and the data can only be obtained on cloudless days. Therefore, a very small number of collocated samples could be achieved in our experiment period. Furthermore, previous studies have shown that the correlation coefficient between MODIS and AERONET AOD is only from 0.505 to 0.75 in the coastal area [65]. Therefore, we will try to use the multi-spectral measurement of ground-based solar radiometer for further analysis in our future work.

**Author Contributions:** T.L. and S.C. (Shengcheng Cui) conducted the experiments and conceived and designed the contents of this paper; X.M., K.Y., G.S. and X.L. contributed to the development of the analysis programs; N.L. performed data visualization as well as formal analysis and wrote the manuscript; T.L., S.C. (Shengcheng Cui) and S.C. (Shunping Chen) reviewed and edited the manuscript. All authors have read and agreed to the published version of the manuscript.

**Funding:** This research has been supported by General Program of the National Natural Science Foundation of China (41875041) and Natural Science Foundation of Anhui Province (2008085J19); Advanced Laser Technology Laboratory of Anhui Province's Foundation (AHL2021QN01); HFIPS Director's Foundation (YZJJ2022QN06); the National Key Research and Development Program of China (2018YFC0213101).

**Data Availability Statement:** Data underlying the results presented in this paper are not publicly available at this time but may be obtained from the authors upon reasonable request.

**Acknowledgments:** We would like to thank all the observers who worked hard on the field campaign during this experiment, the anonymous reviewers for their useful comments, and the editors who provided assistance during the revision, all of them are important in improving this manuscript.

**Conflicts of Interest:** The authors declare no conflict of interest.

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
