# Peer review of "Characteristics of Aerosol Extinction Hygroscopic Growth in the Typical Coastal City of Qingdao, China"

_remotesensing, doi:10.3390/rs14246288_

Round 1

Reviewer 1 Report

This paper addresses an important problem - hygroscopic growth (HG) on the aerosol extinction – for coastal environment. The authors propose an improved retrieval algorithm for aerosol extinction HG factor by taking aerosol type and particle number concentration into account. With long-term field measurements in Qingdao, the authors use an aerosol backward trajectory tracing model to classify the aerosol types into land source, sea source, and/or the mixed. Finally, the HG characteristics of different aerosols types under different seasons were obtained.

Overall, the article is well organized and is in good presentation. However, there are some minor issues needing to be improved: 

1.     The title seems to be more suitable as “Characteristics of aerosol extinction hygroscopic growth in the typical coastal city of Qingdao, China”;

2.     Page 1, the abstract needs to be rewritten to highlight the work with some necessary quantitative results obtained using the proposed algorithm.

3.     Give full spellings for all the terminologies at their first appearance, e.g., GDAS, MODIS,...,

while some abbreviations (e.g., PNC) are redefined;

4.     In Subsection 2.1.1, technical specifications of different instruments are suggested to be listed in one table, including measurement range, resolution, accuracy, sampling frequency, and so on;

5.     In Table 3, the authors should give detail values or ranges for both RMSE and R2, to explain which value or range for the classification for aerosol type in different season is rational and correct;

6.     Page 9, “pure Marine,…”should be “pure marine, …”;

7.     Page 10, “… in the analyze of HGFext…”, analyze -> analysis;

8.     Page 13, Figure 8 b1 and c1 -> Figure 8 (b1) and (c1), FIG. 8 b1 -> Figure 8(b1); Please check the full text.

Reviewer 2 Report

The manuscript is written well and it can be accepted after a moderate revision. I have few specific comments as listed below which may improve the scientific contents of the manuscript.

General Comments:

1. Diurnal feature of AOD along with the visibility data is found to be missing in the manuscript. It is one of the important parameters to explain the variability data. Temporal asymmetry in aerosol optical characteristics is strongly significant in clean (high-altitude, remote, continental and coastal) sites. Authors should elaborate more on this section with relevant recent works reported elsewhere.

2. Can you explain how the visibility varied during the day with aerosol particle number concentration or with other available aerosol optical parameters ?

Specific Comments:

1. Line no. 139: Please used a proper reference instead of using ‘Chwala’s research’.

2. Line no. 165-166: The sentence, ‘It is one of the most…products.’ can be remove as you have describe similar description in the previous sentence. Please mention temporal and spatial resolution of data what you have used in the manuscript. The same can be updated for other data sources.

3. Line no. 184-188: Do you meant RH > 50% for δext,wet in relevant to Equations 1 and 2 ? Please elaborate it.

4. Line no. 293-295: Since you have spectral AOD values, you can estimate Angstrom Exponent (AE). You can also examine the three aerosol types from the AOD vs AE scatter plot.

5. Line no. 300-302: The mixing aerosol types can be explained from the AOD vs AE scattered plot.

6. Figure 2: Diurnal feature of visibility data is found maximum during ~ 12:00 - 22:00 LTC and after that it start falling in the Figure 2. Is this pattern same for other months ? From this study you can see diurnal asymmetry/symmetry of visibility data.

7. Figures 4, 7 and 8 : Apparently, there is lack of observational data during 2020-05. Please mention it in the text.
